# Clinical Tools for Rectal Cancer Response Assessment following Neoadjuvant Treatment in the Era of Organ Preservation

**DOI:** 10.3390/cancers15235535

**Published:** 2023-11-22

**Authors:** Amalia J. Stefanou, Sophie Dessureault, Julian Sanchez, Seth Felder

**Affiliations:** Clinical and Pathologic Response to Therapy in Gastrointestinal Oncology, Moffitt Cancer Center, 12902 Magnolia Dr., Tampa, FL 33612, USA; amalia.stefanou@moffitt.org (A.J.S.); sophie.dessureault@moffitt.org (S.D.); julian.sanchez@moffitt.org (J.S.)

**Keywords:** rectal cancer, watch and wait, non-operative management, wait and see, organ preservation

## Abstract

**Simple Summary:**

This article reviews the contemporary clinical modalities for rectal cancer tumor response assessment following neoadjuvant treatment(s), including physical exam, endoscopy, biopsy, biomarkers, and dedicated rectal cancer imaging. Some patients obtaining a complete or near-complete clinical response following neoadjuvant treatment(s) may safely defer or avoid major pelvic surgery.

**Abstract:**

Local tumor response evaluation following neoadjuvant treatment(s) in rectal adenocarcinoma requires a multi-modality approach including physical and endoscopic evaluations, rectal protocoled MRI, and cross-sectional imaging. Clinical tumor response exists on a spectrum from complete clinical response (cCR), defined as the absence of clinical evidence of residual tumor, to near-complete response (nCR), which assumes a significant reduction in tumor burden but with increased uncertainty of residual microscopic disease, to incomplete clinical response (iCR), which incorporates all responses less than nCR that is not progressive disease. This article aims to review the clinical tools currently routinely available to evaluate treatment response and offers a potential management approach based on the extent of local tumor response.

## 1. Introduction

Over the last decades, a significant number of patients diagnosed with non-metastatic locally advanced rectal adenocarcinoma (LARC) have avoided or deferred radical proctectomy with mesorectal excision following neoadjuvant treatment(s). Largely driven by patients wishing to avoid the life-altering sequelae of major pelvic surgery, patients considered suitable for rectal preservation nonoperative management (NOM), or a Watch and Wait (WW) approach, are often cured without the need for a proctectomy. The decision to pursue NOM or a WW approach is based on clinical and radiographic criteria upon completion of neoadjuvant treatment(s). The magnitude of benefit for a patient otherwise facing a permanent colostomy or coloanal anastomosis able to pursue a high-intensity surveillance approach rather than proctectomy is difficult to quantify but clinically significant. Apart from preserving quality of life by avoiding proctectomy, when studied through Markov modeling comparing WW to radical resection (low anterior resection or abdominoperineal resection), WW has been shown to be significantly more cost-effective than radical resection [1].

Some patients with LARC may accept a lower likelihood of cure if able to avoid rectal surgery, though it remains uncertain whether there is potential oncologic harm to a patient later identified with a local regrowth and requiring a deferred total mesorectal excision (TME) [2]. Recent prospective trials testing total neoadjuvant treatment (TNT) approaches have shown that chemoradiation (CRT) followed by consolidation chemotherapy (CCT) results in rectal organ preservation in a surprisingly high number of patients, approaching 50% [3]. Given the encouraging oncologic outcomes increasingly reported, perhaps rectal organ preservation is evolving closer to the ‘norm,’ rather than the exception [3,4], presciently observed in 2013 by Habr-Gama over a decade ago [5]. Mirroring the evolution of anal squamous cell carcinoma treated with neoadjuvant treatment, a radical resection for rectal cancer may be reserved for only those patients with persisting or recurrent local disease.

For these reasons, it is necessary to understand how best to evaluate rectal cancer patients following neoadjuvant therapy. Given the limited ability to predict tumor response to neoadjuvant treatment(s) currently, patients must be thoughtfully reassessed after neoadjuvant treatment(s) [6,7]. Current tools to estimate the extent of tumor response to neoadjuvant treatment(s) include a digital rectal exam (DRE), endoscopic evaluation, and rectal protocoled MRI with T2 weighted and diffusion weighted imaging (DWI), which in combination are able to provide a reliable clinical probability as to the completeness of local tumor response. Still, the well-recognized limitation of these available tools is reflected in the consistently reported local regrowth rate due to the inability to distinguish residual microscopic disease from fibrosis (or, conversely, the unexpected pathologic complete response, ypT0ypN0 confirmed following TME), emphasizing the importance of clear patient understanding and careful clinician-patient discussion when considering both radical resection and WW. The risk of tumor regrowth varies according to the clinician’s accuracy of tumor response extent, neoadjuvant treatment regimen and timing of assessment, but ultimately, determined by tumor biology. Recognizing this, the clinical T-stage estimates a 10% baseline risk of regrowth with an additive 10% for each T-stage increase [8,9]. The risk of regrowth also appears limited to the first three years post-neoadjuvant treatment, with 97.8% of patients remaining disease-free if they are regrowth-free at 3 years [10]. As clinical complete response (cCR) and WW become an increasingly studied endpoints in rectal cancer clinical trials, additional tools such as sensitive biomarkers (i.e., circulating tumor DNA) or computational biologic analysis through radiomics and machine learning technology may augment accuracy in tumor response assessment thereby improving patient selection at the time of restaging.

## 2. Methods

The evidence reviewed in this manuscript is a summation of relevant scientific publications and current practice guidelines. This review provides expert opinion on the evolving literature reporting clinical assessment of rectal cancer in the context of organ preservation and also provides a management decision algorithm for tumor response extent interpretation. We believe this review is timely and of high clinical importance, as rectal cancer management is rapidly changing and clinical decision-making increasingly complex.

## 3. Watch and Wait (WW)

Neoadjuvant treatment sequences and strategies have evolved over the last several decades, with varying permutations of pelvic (chemo) radiation and systemic chemotherapy. The heterogeneity of tumor responsiveness to these treatments demonstrates the major gap in understanding rectal cancer tumor biology [11]. Most patients experience a partial tumor response to neoadjuvant treatment(s); however, a minority of responses are complete, i.e., without residual viable cancer cells, ypT0ypN0 [12].

Treatment intensification or escalation in the form of radiation dose boosting, novel radiosensitizers applied during long-course chemoradiation, or multi-agent systemic chemotherapy regimens are active areas of research. The intent of treatment escalation is twofold: to improve long-term oncologic prognosis and maximize the likelihood of a pathologic complete response (pCR). On the other hand, the PROSPECT trial studied the de-escalation of pelvic radiation in the hope of avoiding its related toxicity within the overarching tri-modality treatment paradigm of LARC [13]. Schrag et al. reported that the majority of presumed lower-risk rectal cancers do not benefit from radiation when compared to neoadjuvant chemotherapy alone followed by mesorectal excision. Balancing over- and under-treatment to individualize treatment decisions will continue to depend on an improved understanding of the biological heterogeneity affecting rectal cancer responses.

Total neoadjuvant treatment (TNT) strategies are considered a standard treatment paradigm in the US, reserving rectal resection as the last in the sequence of treatment, if necessary. Re-assessment throughout neoadjuvant treatment(s) allows for response-based decision-making rather than empirically prescribed management strategies. The superiority of any given TNT approach over another has yet to be firmly determined; however, all increase the proportion of LARCs reaching a pCR when compared to either neoadjuvant (chemo) radiation or chemotherapy alone prior to resection. The rate of pCR is relatively consistent across published cohorts receiving TNT and ranges between 25% and 38% [12,14,15]. Although rates of cCR are unknown across unselected populations, applying the pCR rate reported from randomized and population studies, the cCR rate is assumed to be similar to the pCR rate [16]. Despite the challenges of accurately identifying a pCR clinically rather than histologically, it is well recognized that radical resection confers no additional oncologic benefit to confirm a pCR histologically.

Which patients should receive treatment intensification (or de-intensification) and for what intent [14,17]? Efforts to personalize treatment(s) by adding or withholding components of the historical tri-modality standard of care paradigms have provided more options for patients but also increased the decision-making complexities, emphasizing the need for clear discussion regarding the uncertainties of over- and under-treatment. Should the priority be the avoidance of radical rectal surgery or focus on the oncologic metrics most commonly applied across cancer patients, such as local failure, disease-free survival, and overall survival? These highly relevant patient endpoints hinge on critical limitations. First, the extent of tumor regression, from minimal regression to complete response, following neoadjuvant treatment remains unpredictable. Second, and most relevant to this review, the current clinically available tools to accurately measure the extent of tumor response to neoadjuvant therapy remain imperfect. A post-treatment scar may contain scattered viable tumor cells not apparent on a clinical, endoscopic, or radiographic exam, falsely assuming the tumor has been entirely eradicated, potentially resulting in a worse oncologic outcome by deferring potentially curative surgery until gross disease is recognized.

## 4. Why Patient Selection Remains Imperfect

Assessing tumor response to neoadjuvant treatment(s) is ultimately subjective and suggests the probability of the presence or absence of a residual tumor. Because a cCR does not necessarily equate to a pCR, and similarly, a patient with a pCR may not satisfy clinical criteria for a cCR, WW must be cautiously approached for each patient. Patients pursuing WW should be closely monitored with CEA, clinical DRE, endoscopic examination, and imaging, including rectal protocoled MRI, at regularly defined, frequent intervals following receipt of neoadjuvant treatment(s). The factor of time from cCR designation heavily influences the conditional risk for a patient, with over 90% of local regrowth occurring within the first 2 years and over 98% within 3 years [8,18]. 

A fundamental limitation of WW is imperfect patient selection. Because DRE, endoscopic inspection of the luminal mucosa, and functional rectal MRI sequences cannot alone detect cancer cells remaining within the bowel wall and mesorectal lymph nodes, clinicians must understand the findings from these evaluations as an association, or clinical probability, of the pathologic gold standard (i.e., pCR). 

Local regrowth following an apparent clinical complete response (cCR) pursuing WW surveillance ranges between 15 and 40% [3]. These moderately high rates of local regrowth indicate suboptimal or inadequate selection criteria from which to select patients. The prospect of patient harm by pursuing WW with unrecognized persistent disease resulting in a potentially increased risk of synchronous or metachronous distant failure at the time of regrowth has tempered the enthusiasm for rectal organ preservation [19,20,21]. These risks remain partially understood since regrowth alone is a marker of unfavorable biology and could explain the association with metastases [22]. Earlier studies assessing the clinical and pathologic concordance of tumor response was prior to standardization of response definitions and did not incorporate functional rectal MRI restaging. The timing of tumor response evaluation and the interval of time from receipt of treatment(s) also strongly influences tumor response interpretation and may capture a ‘snapshot’ of a tumor evolving from partial to complete over an extended period of time [23,24]. Regardless, the relatively high rates of local regrowth highlight the challenges associated with post-neoadjuvant restaging, even when patients are managed in centers with expertise in rectal organ preservation applying strict criteria for selection of WW and follow-up guidelines.

## 5. Rectal Tumor Response Assessment

As experiences with WW mature, the combination of clinical, endoscopic, and rectal MRI assessment remains the most accurate modality for the identification of tumor response after TNT(s) [25,26,27,28,29]. Figure 1 describes a suggested pathway for the evaluation of patients after receiving neoadjuvant treatment(s) and the evaluation of local response. While some patients will have an obviously incomplete response and require TME, others will have complete or near-complete responses. Table 1 summarizes findings seen on endoscopy and MRI T2weighted DWI imaging when assessing for cCR, nCR, and iCR. [28,30] Restaging assessment should typically occur between 6 and 12 weeks following the completion of neoadjuvant treatment(s) [30]. While there is still no clear time ‘cut off’ for evaluation, re-assessment following a short interval up to 12 weeks later when an initial nCR is identified can be pursued to confirm improvement or stability, with likely low risk of disease progression. Up to 90% of patients with a near-complete response (nCR) at the 8–10 week re-assessment time point may reach a cCR 6–12 weeks later, potentially warranting a second interval evaluation prior to pursuing TME [22,31]. The frequency and duration of surveillance for a patient’s designated cCR (or stable or improving nCR) is unknown; however, since ~98% of local regrowth occurs within the first 3 years, a general framework should include frequent clinical exams with endoscopy every 3 to 4 months, at least biannual rectal MRI, and cross-sectional imaging at least yearly for surveillance, with the frequency de-escalating after 2 to 3 years of stability and continuing out to at least 5 years [3,29].

## 6. Biomarkers

As part of the usual rectal surveillance, CEA (carcinoembryonic antigen) is monitored post-receipt of neoadjuvant treatment(s) and trended for patients following a WW approach. The frequency of checking CEA is usually paired with each endoscopy, initially every 3 to 4 months after clinically establishing a cCR (or nCR). There is no clear correlation between the pre-treatment CEA cut-off value and the probability of a favorable tumor response to neoadjuvant treatment(s). Nor is there consistent evidence that post-neoadjuvant CEA levels predict the extent of tumor regression, specifically pCR or non-pCR [32,33]. The serum protein tumor marker CA 19-9 has not been reliably associated with the prediction of tumor response. Hematologic markers, such as thrombocyte and hemoglobin levels, leukocytes, and inflammatory markers, such as albumin and C-reactive protein, as well as leukocyte levels or ratios, have been unable to consistently predict a pCR. 

There are no biomarkers available for clinical use with the accuracy needed to reliably predict or identify pCR or cCR [34]. Molecular or nucleic acid biomarkers, either tissue- or blood-based, such as cell-free or circulating tumor DNA and microRNA (miRNA), may have the potential to predict or measure response to treatment at an earlier time point with sufficient sensitivity and specificity, although to date they remain experimental.

## 7. Digital Rectal Exam

Tactile examination by digital rectal exam (DRE) may detect subtle irregularities not apparent endoscopically or radiographically, yet suspicious for residual tumors. DRE is only possible for lower rectal cancers and is subjective to the examiner. The tumor may not be in reach of the examiner’s reach, particularly in patients with a long anal canal. When assessing tumor response, a cCR is satisfied when the mucosa is regular and smooth fibrosis or normal mucosa is palpated, although thickening and extra-rectal fibrosis are commonly appreciated. A near-complete response (nCR) may have smooth induration or minor mucosal abnormalities or irregularities, including a non-critical or clinically asymptomatic stenosis. In addition, abnormalities on DRE do not preclude a pCR [35]. In a study reported by Guillem et al., DRE correctly identified 21% of patients with a confirmed pCR on post-treatment DRE as a complete clinical response. The extent of pathological downstaging was underestimated in almost 80% of the patients. Other series incorporating DRE into post-neoadjuvant tumor response evaluation are limited and may not adequately represent the utility or reliability of tactile examination [36,37]. Still, DRE is easily applied in conjunction with endoscopy and serves as an initial tool for rectal tumor response assessment.

## 8. Endoscopy

Endoscopy remains the primary modality to assess the degree of rectal tumor response to neoadjuvant treatment(s). Since morphologic changes following neoadjuvant treatment(s) visualized at the time of endoscopic assessment presumedly represent varying degrees of histologic regression, the endoscopic exam is often considered the most reliable method to evaluate the completeness of tumor response. However, a lack of simple and reproducible endoscopic criteria and variability in the perception of endoscopic features characterizing tumor response limit optimal patient selection for WW [38,39]. Although over 90% of tumor regrowth occurs endoluminal and endoscopically detectable, mesorectal assessment for residual nodal disease still requires multi-modal evaluation, including MR imaging. Primary tumor regression with persistence of a pathologic nodal disease is, fortunately, uncommon [40].

Endoscopic evaluation provides a magnified view of the mucosa, allowing a detailed inspection of post-treatment mucosal changes. These findings, including edema, proctitis, stenosis, and reduced elasticity on insufflation, along with other visual characteristics and cues, suggest the extent of tumor response to neoadjuvant treatments. Diagnostic accuracy may be improved by ascribing the presence or absence of reproducible visual features and patterns. The Sao Paulo group was the first to provide formal criteria for classifying endoscopic findings into a binary treatment response assessment—complete or incomplete clinical tumor response [41]. Currently, a three-tiered expert consensus schema assessment includes complete clinical response (cCR), near-complete clinical response (near CR), and incomplete clinical response (iCR). The endoscopic response descriptions include: (1) flat, white scar; (2) telangiectasia; (3) absence/presence of ulceration; (4) absence/presence of nodularity; (5) small mucosal nodules; (6) superficial ulceration; (7) mild persisting erythema of the scar; and (8) visible tumor These are described in Table 1 and Table 2. The correlation between individual endoscopic features and the degree of tumor response following neoadjuvant treatment(s) has not yet been rigorously studied. Advanced endoscopic technology, including narrow-band imaging, confocal laser endomicroscopy, magnifying chromoendoscopy, and machine learning, remain active areas of research to augment endoscopic tumor response assessment [42,43,44].

Residual mucosal abnormalities (RMA) describe mucosal irregularities present endoscopically or on surgical pathologic specimens after neoadjuvant treatment. Several retrospective series have attempted to correlate RMA to the final pathologic stage [45,46]. These investigations showed that most patients fulfilling cCR criteria (using Habr Gama’s 2010 criteria) were correctly correlated to a pCR. However, many patients without a cCR secondary to RMA did not contain histologic residual cancer within the specimen. The authors concluded that examination of the rectal mucosa poorly correlates with pathological responses. Habr-Gama and Perez subsequently emphasized the time-treatment dynamic relationship of tumor response and endoluminal assessment, while also highlighting that 9 of 10 patients would have been correctly identified as a pCR based on endoscopic criteria, favoring the tendency of clinicians to still err on the side of caution when interpreting the completeness of tumor response [45,47].

The application of pre-specified endoscopic criteria has been associated with improved objectivity in endoscopic assessment and, consequently, accurate tumor response degree classification [38,39]. Pre-defined endoscopic features were highly associated with the degree of tumor response across surgeons with varying WW expertise in a retrospective series from Memorial Sloan Kettering Cancer Center [38]. The Maastricht group reported an endoscopic sensitivity of 72% to 90%, applying pre-defined criteria to accurately designate a definite or probable complete response. Of patients considered a definite or probable complete responder, 63% to 78% truly had a pCR or were free of local regrowth for at least 2 years (positive predictive value) [39]. Similar to Habr-Gama’s observations, surgeons were more likely to endoscopically identify complete tumor responses rather than incomplete (including near-complete). This may represent surgeons erring on the side of caution, concerned with missing persistent disease or inadequately treating patients with potential residual tumors. 

Since a considerable proportion of patients may reach a nCR at initial endoscopic restaging, the question of whether to observe, pursue a biopsy, locally excise, or radically resect is less clear but must be considered [48]. A near-complete response encompasses a spectrum of responses ranging from a small but clear residual tumor to one that is highly likely to be a complete response but does not fit the strict criteria. Short interval re-evaluation in 6 to 12 weeks appears safe and helpful to clarify luminal stability or changes. Whether or not it is reasonable to cautiously observe when equivocal findings on rectal examination, endoscopy, or MRI occur depends on the harm-benefit balance of the individual patient [22,49].

## 9. Endoscopic Biopsy

A routine endoscopic biopsy should not be performed, but rather selectively. Endoscopic forceps allow for a limited superficial biopsy and carry a high risk of false-negative sampling. A biopsy may fail to detect persistent malignancy in 25% of cT2 tumors and up to 40% of cT3/T4 tumors [50,51]. This may be due to persistent cancer cells present at the invasive front rather than the superficial mucosal layer. This common pattern of tumor regression contains scattered cancer cells admixed with fibrosis deeper within the bowel wall. A benign biopsy result may give false confidence, resulting in observation rather than oncologic resection for residual viable disease. For this reason, a full-thickness biopsy including muscularis propria has been proposed as a diagnostic and potentially therapeutic approach to provide accurate sampling and therefore pathologic T-staging.

A biopsy is most useful when suspicion is high for viable disease to confirm its presence pathologically (i.e., ruling in disease). Van der Sande et al. reported a significant change in the endoscopic confidence level of tumor response extent following endoscopic biopsy in 4% to 13% of patients, suggesting the uncertain but likely limited role of endoscopic biopsy in determining the completeness of tumor response [39]. In cases where there is discordance between the digital exam and/or imaging and endoscopic findings, luminal stenosis or stricture, mucosal features consistent with a nCR (i.e., pale mucosal nodules, superficial ulceration, or erythema of the scar), or patient refusal to proceed with surgical intervention without cancer confirmation, endoscopic biopsy is reasonable to selectively perform. Although a negative biopsy does not rule out disease, a 2 to 3 month time interval of surveillance should allow the clinician to discern whether the treated tumor is a stable scar or regrowing after an apparent complete or near-complete response.

## 10. Local Excision (Full- or Partial-Thickness Local Excision)

A potential alternative to the shortcomings of a superficial endoscopic biopsy is full-thickness rectal wall local excision (FTLE), allowing the pathologic T-stage to be confirmed. This is especially helpful when there is an equivocal clinical and radiologic interpretation of the degree of tumor response. In this way, FTLE may overcome the inherent subjectivity of clinical, endoscopic, and radiologic assessments, allow the selection of patients for WW, and provide tailored surveillance, particularly when a near clinical complete response is present. Currently, neither the expert consensus of the NCCN nor the ASCRS Clinical Practice Guidelines consider LE an acceptable therapeutic option for a fit patient with LARC pursuing a rectal organ preservation strategy [30,52].

In addition to the inability of FTLE to adequately sample mesorectal nodes, the phenomenon of non-uniform ‘tumor scatter’ beyond the visible residual endoluminal mucosal abnormality has been posed as a limitation when determining local excision lateral margins [53,54,55,56]. Hayden et al. showed tumor cells extending up to 3 cm from the lateral aspect of the residual mucosal abnormality, and Smith et al. reported tumor scatter in 71% of patients, though the maximal distance of residual cancer beyond the RMA in their series was 9 mm, still cautioning that lateral spread under normal mucosa is a common finding in patients after neoadjuvant radiation [57]. 

There are clear benefits to obtaining accurate diagnostic pathologic data; however, FTLE is not widely pursued in part due to the risk of severe postoperative toxicities. Significant wound healing morbidity has been associated with FTLE in the irradiated rectum and has tempered its application as a diagnostic and/or therapeutic management tool [58]. Perez et al. reported a 56% immediate complication rate, 61% experiencing wound dehiscence, and a 43% hospital readmission rate due to severe rectal pain [54]. It is difficult to interpret the morbidity of any patient cohort undergoing FTLE since the extent of perirectal fat, the rectal circumference of excision, proximity to the functional anal canal and anterior peritoneal reflection, anatomic position within the rectum (anterior, posterior), and pre-existing anorectal function influence the morbidity profile and risk. Favorable perioperative outcomes following FTLE have also been reported, so FTLE should not be universally relegated to rectal cancer restaging and may be selectively used as definitive management in highly selected patient [59].

Not only can wound healing complications compound difficulties in subsequent rectal interpretation during surveillance, but FTLE may also compromise the quality of the mesorectal excision specimen’s integrity if radical surgery is later recommended following pathologic T-stage confirmation. Patients recommended to undergo short interval total mesorectal excision following FTLE pathologically confirming more than limited disease or containing unexpectedly higher risk histology (yp > T1, poorly differentiated, lymphovascular invasion, R1 margin) have been shown to have inferior quality of mesorectal specimens, higher rates of non-sphincter sparing resections, and greater perioperative complications [60,61,62,63].

When highly suspected to fulfill criteria for a cCR, FTLE should not be routinely performed to confirm ypT0 pathologically due to the risks for acute and long-term anorectal and bowel impairment when compared to patients observed with a sustained cCR and followed by WW [64]. FTLE might be reserved for patients with an nCR, balancing the benefit of histologic confirmation of ypT status to best determine management with the risks of the local operation [65]. The role of advanced endoscopic interventions such as EMR or ESD compared to surgical excision is currently not yet known; however, the expected fibrosis of the rectal wall may reduce the feasibility of an endoscopic approach to reliably ascertain the extent of mural residual disease and guide management.

## 11. Imaging

Radiographic local rectal tumor response assessment is currently best performed with rectal protocoled MRI and is the currently accepted standard of care [66]. Rectal MRI with T2 and diffusion weighted imaging sequences provides important data estimating the degree of tumor response to neoadjuvant treatment(s), mesorectal and extra-mesorectal lymph node status, and relationship to the mesorectal fascia [67,68]. Specific findings on post-treatment MRI compared to pre-treatment are well described and assist in understanding tumor response. [29]. Table 3 demonstrates examples of possible radiographic response. Favorable radiographic post-treatment fibrotic changes to the tumor and surrounding tissues appear darker in intensity and homogenous in signal. The addition of diffusion weight imaging (DWI) helps to differentiate between fibrosis and high cellularity, potentially reflecting viable disease and improving accuracy compared to T2 weighted tumor response interpretation alone [65]. T2-weighted pre-treatment rectal tumors display intermediate T2-weighted signal intensity compared to the muscularis propria. A complete response on MRI refers to a normal-appearing rectal wall with fibrosis reflected as a dark T2 without an intermediate signal and a lack of restricted diffusion. A near-complete response is designated when there is mostly a dark homogenous T2 signal with some remaining intermediate T2 along with focal or minimally restricted diffusion.

Table 3 MR **Imaging Regression Schema**: For each type of response, we see pre-therapy imaging to the left and post-treatment to the right.

**Complete Clinical Response (cCR)**: cCR highlights a complete response through imaging. For T2 response, cCR has a dark T2 signal, which is well demonstrated on post-therapy images (arrow). Lymph nodes with a cCR show complete or near-complete resolution, which is demonstrated by both cases (arrow). For DWI, cCR displays no signal on high-B value DWI, and for ADC, cCR highlights no low signals (arrows).

**Near-Complete Clinical Response (nCR)**: nCR shows findings of response to therapy with some concern for a small amount of residual disease. nCR has both a dark and intermediate signal on T2 imaging. This patient had a mid-rectal mass with an intermediate signal (arrow) on pre-contrast imaging that extended beyond the muscularis propria (chevron). On post-therapy imaging, there is a combination of dark and intermediate T2 signals with fibrotic tethering but no clear extension beyond the muscularis propria. For lymph nodes, nCR demonstrates regression (arrows). For DWI and ADC maps, nCR demonstrates minimal to low residual signals (arrows). Note that the signal on DWI post-therapy imaging is difficult to detect, but the persistent intermediate signal on T2w imaging confirms the nCR designation.

**Incomplete Response (iCR)**: iCR demonstrates distinct imaging findings of recurrent/residual disease. For T2 response, iCR shows a predominance of intermediate T2 signal with no dark scar. There has been some reduction in the extent of the lower rectal mass (arrow), but post-treatment imaging illustrates clear intermediate T2w signal with no signs of a dark fibrotic response. The chevron details the right side of the puborectalis sling which abuts the mass but with no clear signs of tumor invasion. Regarding lymph nodes, iCR generally demonstrates no change in imaging features. There has been a decrease in the size of the IMA lymph node (arrow) which suggests some response, but an overall conclusion of iCR is still governed by the T2w and DWI features of the mass. For DWI, iCR displays no loss of signal on high b value DWI (arrow). For ADC, iCR illustrates clear loss of signal which is best demonstrated on pre-therapy and post-therapy images.

Despite improvements in technology and experience interpreting post-treatment rectal MRI, accurately distinguishing fibrosis from viable disease remains a challenge. In a randomized, prospective multi-institutional total neoadjuvant treatment trial, the positive predictive value of MRI interpreting a pathologic complete response was only 40%, despite three highly specialized radiologists centrally reviewing and agreeing on the findings. Conversely, the negative predictive value of identifying residual tumor as determined by restaging MRI and confirmed by pathology reached up to 90% [69]. In a Dutch retrospective cohort, the ymrT/ypT stage concordance following neoadjuvant chemoradiation among ‘expert’ and ‘nonexpert’ radiologists was 48% and 43%, respectively, with only moderate overall accuracy, higher in poorer responders than good responders [70]. Among the subgroup of good responders with predominant pathologic fibrosis after neoadjuvant chemoradiation, the ymrT/ypT stage concordance was 40%, along with 44% overstaged. Based on these results, the Rectal MRI Study group concluded that ymrT-staging is most useful to refine the local tumor stage prior to surgery for patients showing no or limited tumor regression but is unreliable and therefore of less value in well-responding patients with predominant fibrosis [71].

Similar to endoscopic evaluation, this may be in part due to the tendency to err on the side of caution when interpreting any area of abnormalities, including fibrosis, at risk of containing a small volume of viable microscopic disease. These data are consistent with other series, highlighting the need to combine MRI morphologic and DWI response patterns with DRE and endoscopy to most accurately assess the extent of local tumor response to neoadjuvant treatment [26,29,72].

The role of FDG–PET for local evaluation, treatment response, and/or surveillance is not well studied. Combining PET with MRI may overcome the limitations of both modalities; however, microscopic residual disease and post-treatment inflammation still result in diagnostic uncertainty [73].

## 12. Conclusions

Currently, the most accurate strategies to assess the extent of rectal tumor response to neoadjuvant treatment(s) include a combination of digital rectal exam, endoscopy, and MRI imaging utilizing T2 weighted and DWI sequences. Despite ongoing refinements of these tools, there remains a moderately high rate of local regrowth, highlighting the shortcomings of recognizing microscopic residual local disease admixed with treatment-related fibrosis. Advances in sophisticated imaging techniques and computational data processing segmentation, along with increasingly sensitive biomarkers for low-volume microscopic residual disease, may enhance the ability to discriminate between a scar and an incomplete response to neoadjuvant treatment(s) to best select rectal cancer patient for nonoperative management.

## Figures and Tables

**Figure 1 cancers-15-05535-f001:**
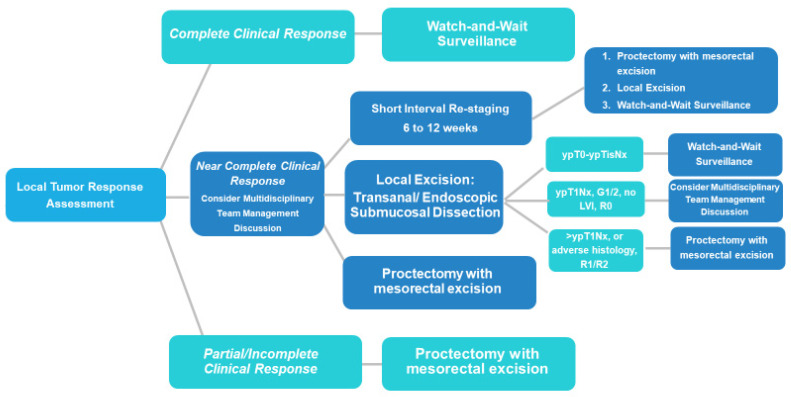
Decision algorithm when evaluating local tumor response extent following neoadjuvant treatment(s).

**Table 1 cancers-15-05535-t001:** Adapted MSKCC criteria of rectal tumor regression schema following neoadjuvant treatment [25,27].

	Complete Clinical Response (cCR)	Near-Complete Clinical Response (nCR)	Incomplete Clinical Response (iCR)
**Endoscopy**	Flat white scar Telangiectasia Absence of ulcers and mucosal nodularity	Small mucosal nodules/minor mucosal irregularities Superficial ulceration Mild, persistent erythema of the scar	Visible tumor
**MRI-T2W**	Only a dark T2 signal; no intermediate signal AND No visible lymph nodes	Mostly dark T2 signal, some intermediate signal AND/OR Partial regression of lymph nodes	More intermediate than a dark T2 signal, no T2 scar AND/OR No regression of lymph nodes
**MRI-DWI**	No visible tumor on the B800–B1000 signal AND/OR Lack of or low signal on the ADC map A uniform linear signal in the wall above the tumor is acceptable	Significant regression of the signal on B800–B1000 AND/OR Minimal or low residual signal on the ADC map	Insignificant regression of signal on B800–B1000 AND/OR Obvious low signal on the ADC map

**Table 2 cancers-15-05535-t002:** Endoscopic mucosal features with examples.

Endoscopic Finding			
**Flat, white scar**	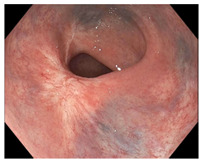	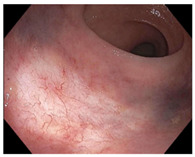	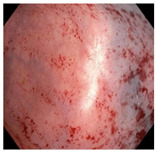
**Telangiectasia**	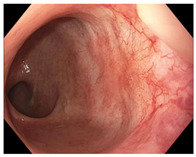	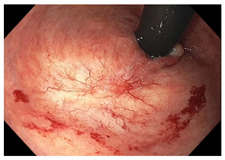	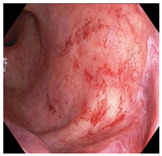
**Small mucosal nodules, minor mucosal irregularities**	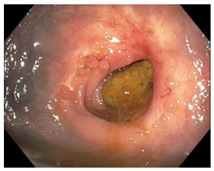	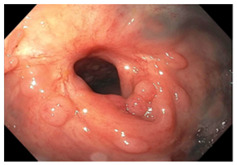	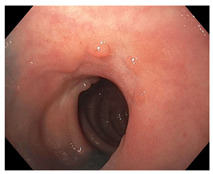
**Superficial ulceration**	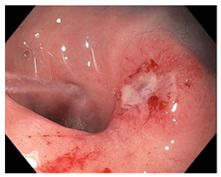	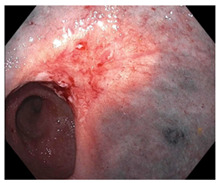	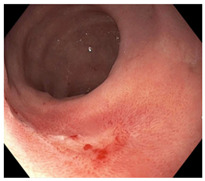
**Persisting erythema of scar**	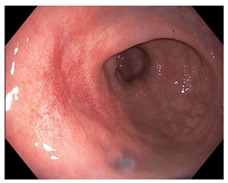	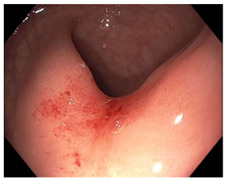	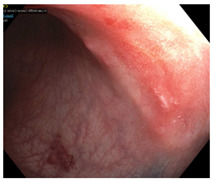

**Table 3 cancers-15-05535-t003:** MRI findings of tumor response following neoadjuvant treatment(s).

Complete Response	Near-Complete Response	Incomplete Response
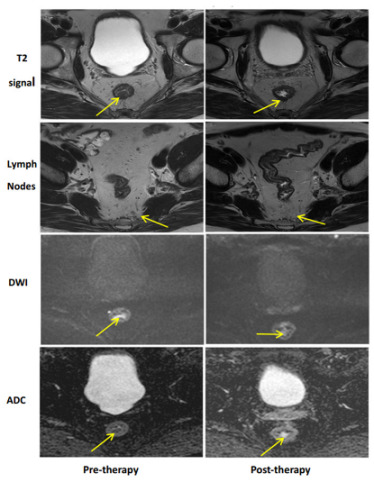	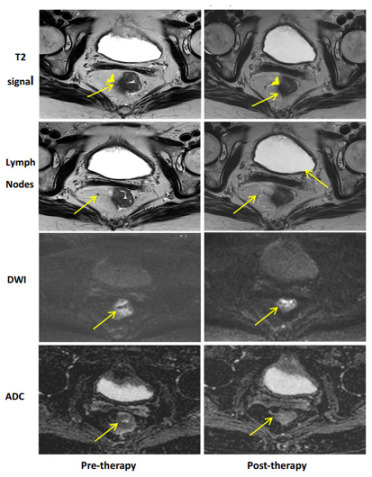	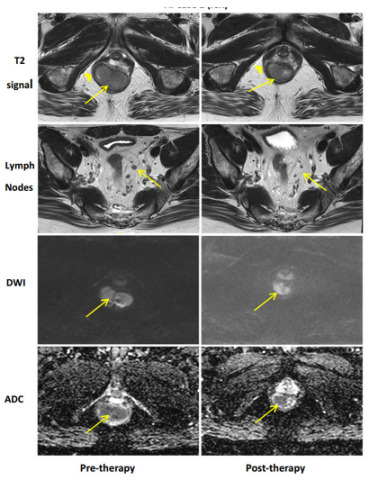

Adapted from Felder et al. 2021 [30].

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
