# Peer review of "Clinical Tools for Rectal Cancer Response Assessment following Neoadjuvant Treatment in the Era of Organ Preservation"

_cancers, 2023, doi:10.3390/cancers15235535_

Round 1

Reviewer 1 Report

Comments and Suggestions for Authors

This is a review article on the available tools for the assessment of the response to neoadjuvant treatment in rectal cancer.

Overall, the article is well-written. However, I would appreciate a more detailed discussion of the following issues:

1.    Timing of response assessment: Particularly, the potential impact of such an assessment during consolidation chemotherapy on the decision to continue or discontinue further chemotherapy.

2.    Sensitivity of cCR diagnosis in relation to the initial T-stage.

3.    Approaches for assessing the response in case of initially enlarged lymph nodes.

Additionally, I recommend including a brief "Methods" section.

Author Response

  1. Timing of response assessment: Particularly, the potential impact of such an assessment during consolidation chemotherapy on the decision to continue or discontinue further chemotherapy.

Thank you. Additional information regarding the timing of evaluation of response and impact or influence on chemotherapy chemotherapy is added.

  1. Sensitivity of cCR diagnosis in relation to the initial T-stage.

Additional data and references have been added.

  1. Approaches for assessing the response in case of initially enlarged lymph nodes.

Thank you. This is included in Table 2, adapted MSKCC regression schema, which describes lymph node response.

Additionally, I recommend including a brief "Methods" section.

This has been addressed.

Reviewer 2 Report

Comments and Suggestions for Authors

The follow-up of patients with rectal cancer after neoadjuvant therapy with the aim of organ preservation is a complex topic. Many questions have not been sufficiently clarified and there is no securely validated procedure to date. The authors discuss the topic in many facets and describe the individual diagnostic options and their combination.  An increasing number of publications with field reports, case studies and so far few prospective or randomised data makes the interpretation and development of valid recommendations difficult. Here, the authors can provide orientation.

Unfortunately, the lack of methodology in the compilation of the literature (keywords, time period of the search) and the selection of the cited studies is problematic. A combination of study methods is also recommended in the summary, a section on studies with combinations is missing in the text.

The discussion of the topics shows that the authors are well versed in the subject and can present the facts in a differentiated way. In my view, the most important topics and data are included and the article provides an overview of the topic. What is missing in particular is a summarising system, e.g. table or decision tree, from which the reader can derive a recommended procedure and adopt it for his practice. With a better presentation of the methodology and the current state of studies and their systematics for the individual sub-items in the text, as well as a clear summary of concrete recommendations for action according to the current state of knowledge, the review could be a valuable orientation aid.

Author Response

Unfortunately, the lack of methodology in the compilation of the literature (keywords, time period of the search) and the selection of the cited studies is problematic. A combination of study methods is also recommended in the summary, a section on studies with combinations is missing in the text.

A methods section is added.

The discussion of the topics shows that the authors are well versed in the subject and can present the facts in a differentiated way. In my view, the most important topics and data are included, and the article provides an overview of the topic. What is missing in particular is a summarizing system, e.g. table or decision tree, from which the reader can derive a recommended procedure and adopt it for his practice. With a better presentation of the methodology and the current state of studies and their systematics for the individual sub-items in the text, as well as a clear summary of concrete recommendations for action according to the current state of knowledge, the review could be a valuable orientation aid.

A Figure with decision algorithm has been created and included.

Reviewer 3 Report

Comments and Suggestions for Authors

This paper is written about clinical tools for rectal cancer response assessment after TNT. It certainly well written.,

I suggest endoscopic evaluation about NBI study cane b introduced and nomogram fot prediction of cCR as well.

MRI figure that showed cCR also be presented for readers

Author Response

This paper is written about clinical tools for rectal cancer response assessment after TNT. It certainly well written.,

I suggest endoscopic evaluation about NBI study cane b introduced and nomogram for prediction of cCR as well.

Thank you, we have included NBI in the text and citations, but not as a currently readily available tool, but as a tool of future potential significance. We aimed to expand in this review on only the most commonly utilized clinical tools.

MRI figure that showed cCR also be presented for readers

We have included MRI images demonstrating degrees of tumor response. 

Round 2

Reviewer 2 Report

Comments and Suggestions for Authors

The article provides a good overview of the topic for clinical users. The methodology now transparently explains the data basis on which the recommendations were developed. The diagram with the decision tree could be a little clearer, but the content is ok and helpful. There are still a few typos in the text.